

# Sensitivity of particle loss to the Kelvin effect in LES of young contrails

Aniket R. Inamdar[1], Alexander D. Naiman[2], Sanjiva K. Lele[1,2], and Mark Z. Jacobson[3]

[1]Dept. of Mech. Engg.
[2]Dept. of Aero. & Astro.
[3]Dept. of Civil & Env. Engg.
[1,2,3]Stanford University, California, USA

*Correspondence to:* Aniket R. Inamdar (`ainamdar@stanford.edu`)

**Abstract.** Different treatments of the Kelvin effect in LES modeling of early contrails are shown to cause variations in the survival rate of ice particles by up to a factor of 4 and in optical depth and mean particle size by up to 50%. The Kelvin effect which varies exponentially with particle size, can reduce or even suppress the impact of other important ambient parameters, such as ice supersaturation, on particle survival rate. Lowering or neglecting the Kelvin effect is shown to substantially alter the

evolution of the ice particle size distribution and delay the onset of particle loss. A strongly Kelvin effect dependent exponential relation between particle survival rate and particle size is shown for high $EI_{soot}$ ($\mathcal{O}(10^{15})$).

## 1 Introduction

While the best estimates for climate impact in terms of global radiative forcing for aviation $CO_2$ emissions ($28 \times 10^{-3} W m^{-2}$) are comparable with those for linear contrails ($11.8 \times 10^{-3} W m^{-2}$) and Aviation Induced Cloudiness(AIC, $33 \times 10^{-3} W m^{-2}$),

the uncertainties in these estimates are substantially larger for contrails and AIC (Lee et al., 2009). Such uncertainties arise due to inadequate or inaccurate representations of contrails in General Circulation Models(GCMs) which stem from a low level of scientific understanding of contrail evolution (Penner et al., 1999). With a consensus on the need to further this scientific understanding, recent work on aircraft contrails has concentrated on the detailed study of young contrails as differences in evolution of young contrails are seen to have lasting effects (Unterstrasser and Gorsch, 2014). There are obvious difficulties

in sufficiently sampling cross-sections of young contrails and measuring the ambient conditions they are formed in. Densely sampled in-situ measurements are indeed few and thus numerical simulation of individual contrails in LES frameworks has become a means of investigation (Chlond, 1998; Lewellen and Lewellen, 1996, 2001; Lewellen et al., 2014; Gierens, 1996; Jensen et al., 1998; Paoli et al., 2003, 2004; Paoli and Garnier, 2005; Huebsch and Lewellen, 2006; Shirgaonkar, 2007; Unterstrasser et al., 2008; Unterstrasser and Gierens, 2010a, b; Unterstrasser and Solch, 2010; Naiman, 2011; Paoli et al., 2013;

Jessberger et al., 2013; Inamdar et al., 2013; Unterstrasser et al., 2014; Unterstrasser and Gorsch, 2014; Unterstrasser, 2014; Inamdar et al., 2014; Picot et al., 2015).

Ice particle transport may be modeled using Lagrangian Particle Tracking (LPT), binned microphysics or bulk microphysics. Previous work using these approaches is summarized in Paoli and Shariff (2016). Within the simulation approaches, those



with LPT are considered to provide more plausible results (Unterstrasser and Solch, 2010), although numerical schemes, initialization procedures and model assumptions vary. Numerically observed sensitivities of particle survival rates, mean size and mean optical extinction to parameters such as $EI_{soot}$ and ice supersaturation($RHi$) reported in the literature (Picot et al., 2015; Unterstrasser, 2014; Naiman et al., 2011) have discrepancies that can be large. In Unterstrasser (2014)(their "Gaussian"

cases in Fig. 13 and $RHi = 130\%$ in Fig. 9), they observe 40% to 70% survival rates for $RHi$ ranging from 120% to 140% and 50% to 90% survival rates for a reduction in $EI_{soot}$ by a factor of 10. In Picot et al. (2015) (their cases 2 and 4 with Kelvin effect in Fig. 19) a change in $RHi$ from 110% to 130% increases survival rate from 65% to near 100% while Naiman et al. (2011) (their Figs. 13,14) see negligible sensitivity of survival rates to $RHi$ ranging from 110% to 130% and an increase of survival rate from 20% to 90% due to a decrease in $EI_{soot}$ by a factor of 10. In Inamdar et al. (2013) and Lewellen et al.

(2014), neglecting the Kelvin effect has a dramatic order-of-magnitude impact on particle survival rates for both young and aged contrails. Here we address the important question of whether the observed discrepancies are due to modeling assumptions and differences in simulation initializations.

The Kelvin effect raises the apparent vapor saturation pressure over a curved ice surface relative to a flat one (Paoli and Shariff, 2016; Jacobson, 2005; Lewellen et al., 2014), for a given temperature, $T$, as follows:

$$15 \quad p_{sat}^{kelvin} = p_{sat}^0 \underbrace{\exp\left(\frac{a_k}{r_p}\right)}_{\Phi(r,a_k(\sigma,T))} \quad \Big| \quad a_k = \frac{2\sigma M_{H_2O}}{RT\rho_p} \tag{1}$$

where $r_p$ is the radius of curvature of the surface, $\sigma$ is the ice-vapor surface tension, $M_{H_2O}$ is the molecular mass of water, $R$ is the universal gas constant, and $\rho_p$ is the density of ice. In Naiman (2011), Naiman et al. (2011), Inamdar et al. (2013), Inamdar et al. (2014), Inamdar et al. (in progress) the parameter $a_k$ was explicitly calculated as a function of $T$ but in other studies the Kelvin effect is neglected (Paoli et al., 2013; Paugam et al., 2010; Paoli et al., 2004) or a fixed order-of-magnitude value of $a_k$

is used (Picot et al., 2015). It is unclear to the authors exactly how the Kelvin effect is modeled in (Solch and Karcher, 2010) and studies employing the same approach (Unterstrasser, 2014; Unterstrasser and Gorsch, 2014; Unterstrasser et al., 2014). Given the wide range of sensitivities of particle survival to $EI_{soot}$ and $RHi$ reported in the above-mentioned studies that use different modeling approaches, it is useful to understand how various treatments of the Kelvin effect can affect properties of young contrails, primarily particle survival rates during the vortex phase.

Exclusion of jet exhaust enthalpy is a common modeling assumption (Unterstrasser, 2014; Unterstrasser and Gorsch, 2014; Unterstrasser et al., 2014; Jessberger et al., 2013; Unterstrasser and Solch, 2010) as it is expected to not have any persistent impact (Schumann, 2012)(their Appendix A6). Keeping in mind that several in-situ measurements provide data for young contrails (Febvre et al., 2009; Voigt et al., 2010; Gayet et al., 2012) and that Unterstrasser (2014)(their "Gaussian" cases in Fig. 13) observes a delayed onset of particle loss ($\sim 40s$) as compared to Naiman et al. (2011)(their Fig. 13) and Inamdar

et al. (2013) (their case "SP" in Fig. 7) ($\sim 20s$), it is necessary to examine the impact of this assumption on the LES of young contrails.



## 2 LES framework, ice microphysics and simulation cases

The LES framework and Ice Microphysics are identical to Naiman (2011) and Naiman et al. (2011) and hence are not detailed here. We perform 12 simulations up to a contrail age of 600 s. Ambient conditions listed in Table 1 were chosen as they are representative of a cruise altitude of 10.5 km (Wilkerson et al., 2010) and are conducive to formation of persistent contrails.

Aircraft parameters that remain constant across simulations are also listed in Table 1. The grid sizes and resolutions are as in Naiman et al. (2011)(their Table 5). The sizes and locations of jet exhaust plumes and vortex cores in the domain are also listed in Table 1, where $\Delta x_{jet/vort}$ is the initial span-wise separation between the pairs of jet exhaust plumes or vortices and $\Delta y_{jet-vort}$ is the initial vertical separation between the jets and vortices, in units of wingspans.

In Table 2, we have listed the simulations performed. We are able to isolate the impact of exhaust enthalpy, $EI_{soot}$, $RHi$ and

10 modeling of the Kelvin effect. Cases with $EI_{soot} = 10^{14}$ and no Kelvin effect are not performed as they are expected to have a near 100% survival rate (Picot et al., 2015). For the aircraft parameters and the $EI_{soot}$ values considered ($\{10^{15}, 10^{14}\}\#kg^{-1}$), the initial ice particles is $\{5.64 \times 10^{12}, 5.64 \times 10^{11}\}\#m^{-1}$.

Unterstrasser (2014) acknowledges that a uniform initialization of ice particles centered around the vortex core is unrealistic and considers the impact of non-uniform (specifically "Gaussian") initialization of ice particle density away from vortex cores

inline with Naiman et al. (2011) but does not include jet exhaust heat. There they observe a substantial difference in the survival rates between their uniform and Gaussian initializations for low and high relative humidities w.r.t ice and hence, we have not considered uniform initialization of ice particles.

The size distribution is initialized to be unimodal. Previous studies (Unterstrasser, 2014; Lewellen et al., 2014) have examined the impact of varying initial size distribution but until measurements of particle sizes in contrails only a few seconds

old become available this "irreducible uncertainty" (Unterstrasser, 2014) in the "true" initial size distribution will persist in simulation efforts. Hence, we have not varied the initial size distribution.

For cases with a temperature dependent Kelvin effect, we choose $\sigma = 0.107\ Jm^{-2}$ (range: 0.100-0.111 $Jm^{-2}$ (Pruppacher and Klett, 2010)) and with spherical ice particles $\rho_p = 917\ kgm^{-3}$. In Table 1, the range of values of $a_k$ for $\sigma \in \{0.100, 0.107, 0.111\}\ Jm^{-2}$ and $T \in \{205, 219.9, 225\}\ K$ are listed and are comparable to Lewellen et al. (2014).

## 25 3 Results

In this section, we show the results of our simulations and highlight the impact of the Kelvin effect treatment on particle survival rates, mean size and mean extinction. We highlight qualitative similarities with earlier studies and reasons for observed differences.

In Fig. 1(a,Top), the evolution of volume-weighted plume temperature in excess of the ambient temperature (henceforth

excess temperature) is shown for all the cases (plume volume is defined by the isosurface of particle concentration = 1 $\#cm^{-3}$). The initial rapid decline (barring cases 'U#') is the rapid cooling due to the warm jet exhaust plume being inducted by the wake vortices. It is clear that the plume temperature remains above the ambient (by $\sim$ 0.3 K) for a majority of the vortex phase and, barring cases 'U#', its variability is less than 0.1 K, *i.e.* the plume temperature in the vortex phase is "constant" (Inamdar





et al., 2016). This behavior will be shown to be a universal phenomenon across variations in aircraft types, ambient conditions and microphysical modeling in an upcoming study by the same authors (Inamdar et al., in progress). Following the destruction of the coherent vortex pair, we see the plume temperature relaxing to the ambient temperature as ambient turbulence mixes the warm plume with the cool ambient. In cases 'U#', we observe the adiabatic heating of the wake as it descends through the

5 stably stratified ambient and the turbulent relaxation to the ambient temperature during the early dispersion phase. Note that the peak plume temperature reached in cases 'U#' is lower than that in cases with jet exhaust heat.

In Fig. 1(b,Top), the Kelvin correction $\Phi(r, a_k(\sigma, T))$ for $a_k(\sigma = 0.107 \ Jm^{-2}, T = 219.9 \ K)$ with bounds using the minimum and maximum values of $a_k(\sigma, T)$ from Table 1 along with $a_k = 10^{-9} \ m$ is plotted as a function of particle radius. It is clear that $\Phi(r, a_k(\sigma, T))$ is a weak function of both $\sigma$ and $T$, but $a_k = 10^{-9} \ m$ results in substantially lower Kelvin correction

for smaller particles($r < 0.1 \ \mu m$). The increase in plume temperature coupled with growth of ice particles results in the reduction of the plume $RHi$ (Naiman et al., 2011; Lewellen et al., 2014). In Fig. 1(b,Bottom) we illustrate the size dependence of excess vapor pressure available for particles for $a_k \in \{\sim 2.29 \times 10^{-9}, 10^{-9}\} \ m$ at $T_{plume} = (219.9 + 0.3) \ K$ for representative reduced plume supersaturations of $RHi_{plume} \in \{105, 101\}\%$. We can see that smaller particles experience apparent subsaturation, the strength of which is strongly influenced by treatment of Kelvin effect. However, the contrail ice mass is insensitive

to both $EI_{soot}$ and the Kelvin effect as seen in Fig. 1(a, Bottom). Thus, larger particles grow at the expense of smaller particles as suggested in Inamdar et al. (2016).

In Fig. 2(Top), the particle survival rate is shown for all the cases. It is important to note that cases with lower Kelvin effect have a delayed onset of particle decay for both high and low $RHi$ values (this effect is prominently seen in cases with high $EI_{soot}$)(Naiman et al., 2011; Inamdar et al., 2013; Unterstrasser, 2014; Lewellen et al., 2014). Also, the delay in onset of

20 decay due to lower $RHi$ is exacerbated due to lower Kelvin effect. The impact of lower Kelvin effect is most prominently felt for higher $EI_{soot}$ as it affects smaller particles (typically $< 0.1 \ \mu m$) and for the same exhaust vapor mass, lower $EI_{soot}$ results in larger ice particles at our simulation initialization (Paoli et al., 2013). Cases with lower Kelvin effect are seen to have higher survival rates and this difference is substantial for high $EI_{soot}$. Qualitatively, the low Kelvin cases with high $EI_{soot}$ are observed to produce similar survival rates and trends w.r.t sensitivity to $RHi$ as seen in Unterstrasser (2014)(their "Gaussian"

cases in Fig. 13) by $\sim 5$min. The $\sim 99\%$ survival rate at 2min. for high $RHi$ and low $EI_{soot}$ is comparable to Picot et al. (2015) (their run 2 in Table 3).

The apparent subsaturation experienced by small particles shows stronger dependence on the treatment of Kelvin effect than to $RHi$ (see Fig. 1(b,Bottom)). It is for this reason that the relative impact of $RHi$ on particle survival is suppressed in high $EI_{soot}$ and high Kelvin cases. In cases that exclude jet exhaust heat, we observe a delay in the onset of particle decay. This

may be due to the lower plume temperature experienced by the ice particles as seen in Fig. 1. But this effect is indeed reduced to insignificant by $\sim 5$ min. As expected, we observe a very high survival rate in cases without Kelvin effect and high $EI_{soot}$.

In Fig. 2(Center, Bottom) evolution of mean particle size(radius) and a log-log plot of particle survival rate vs. mean particle size are shown. As suggested in Inamdar et al. (2016), in case of high $EI_{soot}$, we observe an exponential relation between survival rate and size during the vortex phase, seen in the linearity of the plots in Fig. 2(Bottom). This relationship is only

35 weakly dependent on $RHi$ as the plume temperature remains constant during the vortex phase. But, as the plume temperature





relaxes to the ambient temperature, we observe the plots for high and low $RHi$ begin to diverge. The change in slope as the Kelvin effect is reduced can also be seen. As more particles survive, mean particle size is lowered due to competition for the same vapor mass (Inamdar et al., 2013; Lewellen et al., 2014). The onset of decay is seen to occur at the same particle size, given the $EI_{soot}$ ($\sim 0.75\ \mu m$ for $10^{15}$ and $\sim 2\ \mu m$ for $10^{14}$). Since it takes longer for the mean size to grow to these "critical"

sizes in ambients with lower $RHi$ (see Fig. 3), the onset of decay is delayed in these cases as seen in Fig. 2(Top).

In Fig. 3, we have shown the particle size PDFs at 1 s, 5 s, 15 s, 30 s, 45 s, 60 s, 90 s, 120 s and 300 s. The impact of Kelvin effect on the tail of the size PDF is clearly visible in the presence of large number of small particles for cases with low and no Kelvin effect at later times. We also see that the PDF peaks at lower values for low RHi cases, as is expected. The early PDFs at t = 1 s and t = 5 s are indifferent to the Kelvin effect and qualitatively similar to Karcher and Yu (2009). We note that PDFs

of cases with low $RHi$ take longer to thicken and develop the characteristic size PDF associated with contrails (Naiman et al., 2011; Picot et al., 2015; Lewellen et al., 2014). Since the low RHi cases need longer to develop a tail of small particles, the onset of particle loss is also delayed as compared to cases with high RHi. While this is in contrast to Unterstrasser (2014) and Picot et al. (2015), we strongly suspect this is due to the fact that there the initial size PDF is not unimodal. In Unterstrasser (2014), initializations with thicker size PDFs see an early onset of decay.

Finally, in Fig. 4, the mean optical depth as calculated in Picot et al. (2015) along with the volume-weighted mean extinction (plume volume as defined earlier) is shown for the high $EI_{soot}$ cases only, since there is essentially no impact of Kelvin effect for low $EI_{soot}$. Though, we report for low Kelvin effect and low $EI_{soot}$ a mean optical depth of 0.19 for $RHi$ 130% at 4 min and this value compares well with Picot et al. (2015) (their Fig. 19). Also, we report an optical depth of 0.1 and extinction of 1.5 $km^{-1}$ for low $EI_{soot}$ and $RHi$ of 110% at $t = 100\ s$ which compares reasonably with Jessberger et al. (2013) (their Table

4, unit A319 and $RHi$ 103%) considering our slightly higher $RHi$ and larger aircraft.

In Jessberger et al. (2013) (their Table 5, unit A319, RHi 120%, $EI_{soot} = 10^{15}$), the CoCiP model that does not consider Kelvin effect reports an optical depth of 0.38 and extinction of 2.7 $km^{-1}$ (at $\sim 100$ s and assumed survival rate of 0.5). A similar survival rate ($\sim 0.6$) at the same $EI_{soot}$ is observed for low Kelvin and hence we compare their results with this case where we report an optical depth of 0.45 and extinction of 5 $km^{-1}$ - reasonable considering our larger aircraft size and higher

$RHi$. A number of reported values of optical depth are listed in Paugam et al. (2010) (their Table 3). For a B747 and $RHi$ 130%, the computed mean optical depth (using LES (Unterstrasser and Gierens, 2010a)) up to 35 min is 0.4 which compares well with the value predicted by our high $EI_{soot}$, $RHi$ 130% and low Kelvin, 0.38 (B747 is marginally larger than the aircraft size considered here).

While the impact of modeling of Kelvin effect on optical depth is not negligible for high $EI_{soot}$, it is the most pronounced in

highly supersaturated conditions. We clearly see how lowering the Kelvin effect exacerbates the impact of $RHi$ on the optical depth. The presence of a large number of smaller particles in the low and no Kelvin cases can increase the optical depth by up to 50%.





## 4   Conclusions

Neglect or a simple order-of-magnitude accounting of the Kelvin effect in LES of young contrails may produce substantially higher particle survival rates and also exacerbate the impact of ambient ice supersaturation, especially for high soot emission indices. There is a need for in-situ measurements of young contrails (aged a few seconds up to a few minutes) in weakly and
strongly supersaturated ambients to guide future simulation efforts in this field.

1. The discrepancies in particle survival rates and sensitivities to $RHi$ observed in LES of early contrails using different modeling approaches are primarily due to the different treatment of the Kelvin effect.

2. Particle survival rate and optical depth are the most sensitive to the treatment of the Kelvin effect for high $EI_{soot}$. This sensitivity is reduced to negligible by lowering the $EI_{soot}$ by a factor of 10.

3. In the absence of in-situ data for high $EI_{soot}$ and high $RHi$, it remains an open question as to which modeling approach is more reliable in high soot and high supersaturation regimes, though a temperature dependent treatment of the Kelvin effect seems more plausible.

4. Lowering of the Kelvin effect and exclusion of jet exhaust heat both result in a delayed onset of particle loss, though the effect of exhaust heat is not seen to be persistent.

5. An exponential relation exists in the vortex phase between particle survival rate and mean particle size for high $EI_{soot}$.

6. Onset of particle loss is seen to occur at a "critical" mean particle size that is $EI_{soot}$-dependent. This observation coupled with the slower growth of ice particles in plumes experiencing lower ambient $RHi$ helps explain the delayed onset of particle loss in cases with lower $RHi$.

## 5   Code and data availability

The code and data used in this study is readily available upon request to the corresponding author.

*Author contributions.*  A. R. Inamdar and A. D. Naiman performed the simulations while A. R. Inamdar analyzed the simulation data and composed the manuscript under the supervision of S. K. Lele and M. Z. Jacobson.

*Competing interests.*  The authors declare that they have no conflict of interest.

*Acknowledgements.*  This project is sponsored by the Federal Aviation Administration under the award number DTFAWA-05-D-0006. Tera-
Grid and XSEDE resources LONI, NCSA, KRAKEN and STAMPEDE under grant number TG-CTS080041N were used for this research.





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



**Table 1.** List of Variables

| Ambient Conditions | | | |
|---|---|---|---|
| **Parameter** | **Value** | **Parameter** | **Value** |
| Air Density ($\rho_0$) | 0.387 kg m$^{-3}$ | Relative Humidity | 130%, |
| | | w.r.t Ice (RHi) | 110% |
| Ambient Pressure ($p_0$) | $2.45 \times 10^4$ Pa | Ambient Temperature ($T_0$) | 219.9 K |
| Shear | 0 s$^{-1}$ | Brunt-Väisälä freq. ($N_{bv}$) | 0.01 s$^{-1}$ |
| Ambient Turbulence ($\epsilon$) | $1 \times 10^{-4}$ m$^2$s$^{-3}$ | Turb. Kinetic Energy (k) | $1 \times 10^{-2}$ m$^2$s$^{-2}$ |
| **Aircraft Parameters** | | | |
| **Parameter** | **Value** | **Parameter** | **Value** |
| Wingspan (B) | 50 m | Weight (W) | 136814kg |
| Circulation $\Gamma_0$ | 370 m$^2$s$^{-1}$ | Fuel Burn ($\dot{m}_f$) | $5.64 \times 10^{-3}$ kg m$^{-1}$ |
| Engine Efficiency ($\eta$) | 0.3 | Exhaust Heat | $43 \times 10^6 (1 - \eta)\dot{m}_f$ J m$^{-1}$ |
| **Vortex & Jet Exhaust Initialization Parameters** | | | |
| **Parameter** | **Value** | **Parameter** | **Value** |
| $r_{jet}$ | 7 m | $r_{vort}$ | 3.3 m |
| $\Delta x_{jet}$ | 0.52 | $\Delta x_{vort}$ | $\frac{\pi}{4}$ |
| $\Delta y_{jet-vort}$ | 0.26 | | |

| Range of Values for Kelvin Parameter $a_k (10^{-9} m)$ | | | |
|---|---|---|---|
| | $\sigma = 0.100 Jm^{-2}$ | $\sigma = 0.107 Jm^{-2}$ | $\sigma = 0.111 Jm^{-2}$ |
| $T = 205K$ | 2.30 | 2.47 | 2.56 |
| $T = 219.9K$ | 2.15 | 2.29 | 2.38 |
| $T = 225K$ | 2.10 | 2.25 | 2.33 |



**Table 2.** List of LES Cases

| Case Name | Case Description | Legend |
|---|---|---|
| Baseline (B0) | $EI_{soot} = 10^{15}, RHi = 130\%, a_k = f(T)$ | solid black |
| B1 | $EI_{soot} = 10^{15}, RHi = 110\%, a_k = f(T)$ | dashed black |
| B2 | $EI_{soot} = 10^{15}, RHi = 130\%, a_k = 10^{-9}m$ | solid blue |
| B3 | $EI_{soot} = 10^{15}, RHi = 110\%, a_k = 10^{-9}m$ | dashed blue |
| B4 | $EI_{soot} = 10^{15}, RHi = 130\%, a_k = 0m$ | solid red |
| B5 | $EI_{soot} = 10^{15}, RHi = 110\%, a_k = 0m$ | dashed red |
| L0 | $EI_{soot} = 10^{14}, RHi = 130\%, a_k = f(T)$ | solid green |
| L1 | $EI_{soot} = 10^{14}, RHi = 110\%, a_k = f(T)$ | dashed green |
| L2 | $EI_{soot} = 10^{14}, RHi = 130\%, a_k = 10^{-9}m$ | solid cyan |
| L3 | $EI_{soot} = 10^{14}, RHi = 110\%, a_k = 10^{-9}m$ | dashed cyan |
| U0 | Same as B0, but no jet exhaust enthalpy | solid magenta |
| U1 | Same as B1, but no jet exhaust enthalpy | dashed magenta |





(a) Evolution in Time                    (b) Size Dependence

**Figure 1.** [a] Evolution of (Top) Excess Plume Temperature and (Bottom) Contrail Ice Mass. [b] Size Dependence of (Top) Kelvin correction and (Bottom) Excess Vapor Pressure



(a) $EI_{soot} = 10^{15}$

(b) $EI_{soot} = 10^{14}$

**Figure 2.** (Top) Fraction of surviving particles vs. Time, (Center)Mean Particle Size vs. Time, (Bottom) Fraction of surviving particles vs. Mean particle size







(a) t = 1s

(b) t = 5s

(c) t = 15s

(d) t = 30s

(e) t = 45s

(f) t = 60s

(g) t = 90s

(h) t = 120s

(i) t = 300s

**Figure 3.** PDF of Particle Size Distribution




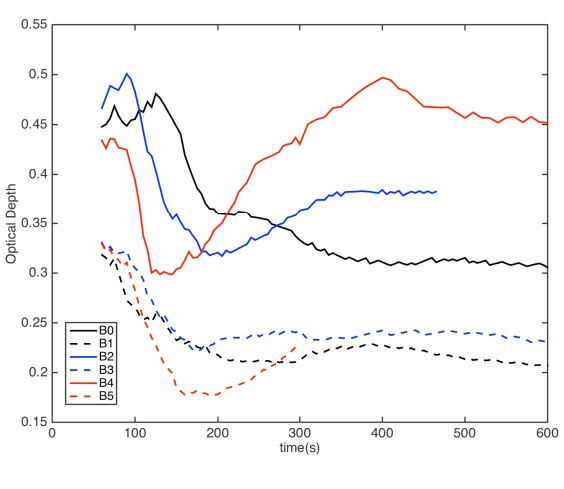
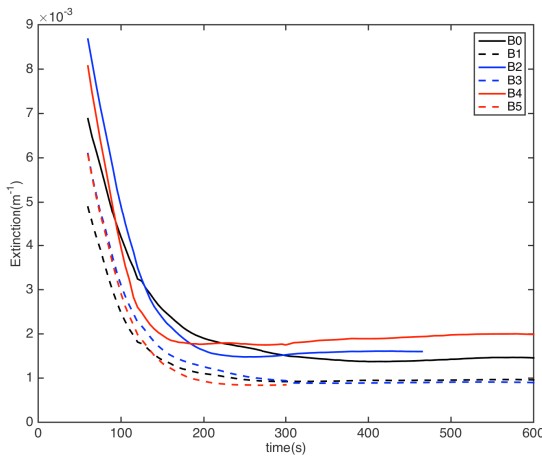

(a) Mean Optical Depth                              (b) Mean Extinction

**Figure 4.** Evolution of mean optical properties