# Peer review of "Sensitivity of particle loss to the Kelvin effect in LES of young contrails"

_Atmospheric Chemistry and Physics, 2016_

## Referee Comment (RC1) · Anonymous Referee #1 · 15 Nov 2016

This is well written and clearly describes a study of how the Kelvin effect can influence the evaporative loss of particles in contrails. The approach and methods are laid out well, and the figures and tables are appropriate.

Two minor comments:

Somewhere in the text (probably page 3, third paragraph or so), the range of particles sizes used as inputs should be listed, and perhaps also in Table 1. This range appears in the figures, but the reader should be explicitly informed as well.

End of page 2 (line 20), it is stated "it is necessary to examine the impact ... [of jet exhaust enthalpy]". Yet in the conclusions, the result of this examination is a minor clause of conclusion 4 ( "though the effect of exhaust heat is not seen to be persistent"). If it is considered necessary and important, the conclusion should be more prominent

and more fully stated.

But my major comment is the following:

While the impact of the Kelvin effect on particle loss is studied here, even the results shown in this paper indicate that the potential impact on initial particle growth is much more significant. Looking at figure 1 b, the steep slopes for particles in the 10 - 100 nm range show that much that is going on is sensitive to the early particles sizes (which start at 10s of nm for aircraft PM emissions). Further, comparing Figure 2 a and b shows that the initial particle number has a larger impact than the variation in Kelvin parameter. The initial size and number is defined by competition for water vapor by the initial condensation nuclei (soot particles). Thus, if the Kelvin effect is important for evaporative loss, it also will affect the initial condensational growth. Figures 1 and 2 strongly suggest that the initial number and size, as determined by Kelvin effect mediated competition for water vapor are much more important than the more subtle effects of Kelvin effect on evaporative loss.

My opinion is that the authors should have noticed this, and extended their study to understand the more important end (the initial size and number) of where the Kelvin effect influences things. I don't take issue with the effect of loss, but the paper gives the strong impression that this is the important consequence, when the results they present suggest otherwise.

At the very least, the potential for even bigger Kelvin effects on initial particle properties (size and number) must be clearly stated. And then state that these big effects on initial properties also affect the loss that they are studying (per Figs 1 and 2) and this must be explained fully. But really, it is my opinion that to be scientifically honest, the authors should go back an include a study of the impact of the Kelvin effect on initial condensational growth and have a more complete package, since I think they have focused on a secondary process which is is strongly influenced by what they have left out.

My recommendation is to reject this paper, and the authors should do the complete study that includes the influence of Kelvin effect on the initial condensational growth. If the editor finds this paper's limited scope acceptable, I would strongly maintain that it should only be accepted if the limitations of leaving out the likely dominance of the Kelvin effect on condensational grown are clearly discussed and the results of this study and how they are used are caveated appropriately. But I would prefer that they go back and explore the broader impact of the Kelvin effect by exploring how it affects initial size and number via condensational growth as well as evaporative loss.

---

## Author Comment (AC1) · 23 Nov 2016

**Interactive comment* on "Sensitivity of particle loss to the Kelvin effect in LES of young contrails" by Aniket R. Inamdar et al.**

Aniket R. Inamdar[1], Alexander D. Naiman[2], Sanjiva K. Lele[1,2], and Mark Z. Jacobson[3]

[1]Dept. of Mech. Engg.
[2]Dept. of Aero. & Astro.
[3]Dept. of Civil & Env. Engg.
[1,2,3]Stanford University, California, USA

*Correspondence to:* Aniket R. Inamdar (`ainamdar@stanford.edu`)

The authors would like to thank Anonymous Referee #1 for his/her comments. The following are the authors' responses to the minor and major comments made by the referee:

*Comment*:

**This is well written and clearly describes a study of how the Kelvin effect can influence the evaporative loss of particles in contrails. The approach and methods are laid out well, and the figures and tables are appropriate.**

*Response*:

Thank You!

**Minor Comments:**

**Range of Particle Sizes**

*Comment*:

**Somewhere in the text (probably page 3, third paragraph or so), the range of particles sizes used as inputs should be listed, and perhaps also in Table 1. This range appears in the figures, but the reader should be explicitly informed as well.**

*Response*:

The initial size of particles is now mentioned in Sec. 2 and is also included in Table 1 of the revised manuscript.

**Impact of Exhaust Enthalpy**

*Comment*:

**End of page 2 (line 20), it is stated "it is necessary to examine the impact ... [of jet exhaust enthalpy]". Yet in the conclusions, the result of this examination is a minor clause of conclusion 4 ( "though the effect of exhaust heat is not seen to be persistent"). If it is considered necessary and important, the conclusion should be more prominent and more fully stated**

*Response*:

Conclusion 4 in the revised manuscript is modified to explain in greater detail the influence of exhaust enthalpy.

**Major Comment:**

*Comment*:

5    **While the impact of the Kelvin effect on particle loss is studied here, even the results shown in this paper indicate that the potential impact on initial particle growth is much more significant. Looking at figure 1 b, the steep slopes for particles in the 10 - 100 nm range show that much that is going on is sensitive to the early particles sizes (which start at 10s of nm for aircraft PM emissions). Further, comparing Figure 2 a and b shows that the initial particle number has a larger impact than the variation in Kelvin parameter. The initial size and number is defined by competition for**

10   **water vapor by the initial condensation nuclei (soot particles). Thus, if the Kelvin effect is important for evaporative loss, it also will affect the initial condensational growth. Figures 1 and 2 strongly suggest that the initial number and size, as determined by Kelvin effect mediated competition for water vapor are much more important than the more subtle effects of Kelvin effect on evaporative loss.**

**My opinion is that the authors should have noticed this, and extended their study to understand the more important**

15   **end (the initial size and number) of where the Kelvin effect influences things. I don't take issue with the effect of loss, but the paper gives the strong impression that this is the important consequence, when the results they present suggest otherwise. At the very least, the potential for even bigger Kelvin effects on initial particle properties (size and number) must be clearly stated. And then state that these big effects on initial properties also affect the loss that they are studying (per Figs 1 and 2) and this must be explained fully. But really, it is my opinion that to be scientifically honest, the authors**

20   **should go back an include a study of the impact of the Kelvin effect on initial condensational growth and have a more complete package, since I think they have focused on a secondary process which is is strongly influenced by what they have left out.**

**My recommendation is to reject this paper, and the authors should do the complete study that includes the influence of Kelvin effect on the initial condensational growth. If the editor finds this paper's limited scope acceptable, I would**

25   **strongly maintain that it should only be accepted if the limitations of leaving out the likely dominance of the Kelvin effect on condensational grown are clearly discussed and the results of this study and how they are used are caveated appropriately. But I would prefer that they go back and explore the broader impact of the Kelvin effect by exploring how it affects initial size and number via condensational growth as well as evaporative loss.**

*Response*:

30   Here, the referee suggests that since we saw a large sensitivity to the Kelvin effect in the vortex phase, we should also expect the same sensitivity, if not more, in the jet phase and examine that phase too. We thank the referee for this insightful comment. However, the jet phase is very unlikely to be as sensitive to the Kelvin effect because the conditions experienced by the plume

are very different in the jet phase than in the vortex phase. The jet phase dynamics are an order of magnitude faster than the vortex phase and the plume experiences strong ice supersaturation in the jet phase as against near-saturation in the vortex phase.

In the present study, we observe that the plume temperature remains nearly constant and warmer than ambient during the vortex phase (our Figure 1(a,Top)) and simultaneously the plume $RHi$ remains only marginally supersaturated (close to 100%) (Naiman et al., 2011; Paoli and Shariff, 2016; Lewellen et al., 2014). At near-saturation, even small changes in the value of $a_k(\sigma, T)$ (our Eqn. 1), can result in dramatic increase in the apparent sub-saturation experienced by small particles (Figure 1(b, Bottom)). This is sustained for the entire period of vortex phase ($\mathcal{O}(10s)$) resulting in large and irreversible loss of small particles.

The jet phase has been extensively studied by Karcher et al. (2015); Wong and Miake-Lye (2010); Karcher and Yu (2009); Yu and Turco (1998) to name a few. In early jet phase, the plume temperature falls rapidly and consequently the RHi increases to very high values (peak $\sim 200\%$) due to the presence of vapor emitted by the engine (see Figure 1(b) in Karcher and Yu (2009)). The Kelvin correction factor, $\boldsymbol{\Phi}(r, a_k(\sigma = 0.107, T = 230))$, for $r \in \{5, 10\}nm$ is calculated to be $\{1.55, 1.25\}$ respectively. We see a $\sim 55\%$ increase in apparent saturation vapor pressure for a 5nm particle. Thus, at such high plume $RHi$ ($\sim 200\%$), particles even of sizes 10's of nm, will still experience supersaturation after Kelvin correction, albeit weakened. In the above example, the 5nm particle at $T = 230K$ and $RHi = 200\%$ will still experience an excess vapor pressure of $\sim 4Pa$ and grow in size.

To demonstrate that even particles of size $\mathcal{O}(10^{-8}m)$ experience positive excess vapor pressure, we have attached a figure plotting the excess vapor pressure over the Kelvin-corrected saturation vapor pressure similar to our Fig. 1(b, Bottom) but with RHi $\in \{150\%, 130\%\}$ and T = 230K. Experiencing a net positive excess vapor pressure, the particle size quickly increase to an almost unimodal size PDF around $\sim 1\mu m$ as seen in Wong and Miake-Lye (2010). Our initialization of particle number and size is consistent with the results of such studies (compare our Figure 3(b) and Figure 2(e) in Karcher and Yu (2009)).

[Figure]

| (a) Near-Saturation | (b) Strong Supersaturation |

**Figure AR 1.** Excess Vapor Pressure - (a) Representative of plume in vortex phase, (b) Representative of plume in early jet phase

We strongly contend that the weakening of the supersaturation due to Kelvin effect in the jet phase will have little impact on the particle number and size. From Figure 1(b) of Karcher and Yu (2009), we infer the time for uptake of vapor emitted by the engine to be $\sim 0.2s$ and a similar time scale can be inferred from Wong and Miake-Lye (2010). Consider an activated soot particle of initial size 10nm, similar to the initialization in Karcher and Yu (2009); Wong and Miake-Lye (2010), experiencing

5    the following externally specified $RHi$ and $T$ variation as inferred from Figure1(b) of Karcher and Yu (2009):

$$RHi = \begin{cases} 200 - 100 \times \frac{\log(\frac{t}{0.1})}{\log(3)} & t \in [0.1, 0.3]s \\ 100 & t \in [0.3, 0.5]s \end{cases} \quad AND \quad T = 235 - 15 \times \frac{\log(\frac{t}{0.1})}{\log(5)} \quad t \in [0.1, 0.5]s \tag{1}$$

Using Eqn. 9 from Naiman et al. (2011) we can estimate the growth of this particle with and without Kelvin effect and this is shown in Figure AR 2. From these plots, we may expect in the jet phase, the initial rapid growth of particles to sizes of $\mathcal{O}(1\mu m)$ due to very high plume $RHi$ to render the Kelvin effect insignificant. This simple analysis leads us to conclude,

10   different modeling of the Kelvin effect may not affect our simulation initialization (a few seconds behind the aircraft) in any significant way. An exact quantification, if necessary, of how different treatments of the Kelvin effect may affect the uptake of vapor in the jet phase may be left to studies equipped with the simulation framework to examine the chemically reactive and compressible dynamics of the early jet phase, which the current simulation framework is not capable of resolving. But, given the above reasoning, we believe this computationally expensive exercise may yield insignificant results.

[Figure]

**Figure AR 2.** Model Jet Phase Ice Growth

To reiterate, different treatment of the Kelvin effect has a large impact on evaporative loss of particles in the vortex phase as the plume experiences $RHi$ close to saturation and nearly constant temperature warmer than the ambient for 10s of seconds. Kelvin effect is very unlikely to have a significant impact on the jet phase as the $RHi$ in the jet phase is substantially higher than saturation and the time scale of vapor uptake in the jet phase is an order of magnitude faster than the time scale of vortex dynamics.

This discussion on the Kelvin effect in the jet phase has been included in the revised manuscript as an appendix.

**References**

Karcher, B. and Yu, F.: Role of aircraft soot emissions in contrail formation, Geophysical Research Letters, 36, 2009.

Karcher, B., Peter, T., Biermann, U. M., and Schumann, U.: The Initial Composition of Jet Condensation Trails, Journal of the Atmospheric Sciences, 53, 3066–3083, 1996.

5  Karcher, B., Burkhardt, U., Bier, A., Bock, L., and Ford, I.: The microphysical pathway to contrail formation, Journal of Geophysical Research, 120, 7893–7927, 2015.

Lewellen, D. C., Meza, O., and Huebsch, W. W.: Persistent Contrails and Contrail Cirrus. Part I: Large-Eddy Simulations from Inception to Demise, Journal of the Atmospheric Sciences, 71, 4399–4419, 2014.

Naiman, A., Lele, S., and Jacobson, M. Z.: Large eddy simulations of contrail development: Sensitivity to initial and ambient conditions over

10  first twenty minutes, Journal of Geophysical Research, 116, 2011.

Naiman, A. D.: Modeling Aircraft Contrails And Emission Plumes For Climate Impacts, Ph.D. thesis, Stanford University, 2011.

Paoli, R. and Shariff, K.: Contrail Modeling and Simulation, The Annual Reviews of Fluid Mechanics, 48, 393–427, 2016.

Shirgaonkar, A. A.: Large Eddy Simulation of Early Stage Aircraft Contrails, Ph.D. thesis, Stanford University, 2007.

Wong, H.-W. and Miake-Lye, R. C.: Parametric studies of contrail ice particle formation in jet regime using microphysical parcel modeling,

15  Atmospheric Chemistry and Physics, 10, 3261–3272, 2010.

Yu, F. and Turco, R. P.: Contrail formationand impactson aerosolpropertiesin aircraft plumes:Effects of fuel sulfur content, Geophysical Research Letters, 25, 313–316, 1998.

---

## Referee Comment (RC2) · Anonymous Referee #2 · 28 Nov 2016

Review of Inamdar et al

**Summary**

The study presents LES of young contrails and focuses mainly on one aspect, i.e. the treatment of the Kelvin effect in the ice microphysical model (which is also reflected in the title). The numerical model has been used before in the study of Naiman et al, 2011. Compared to other LES results, the latter study showed a rather weak sensitivity to ambient relative humidity which appears counter-intuitive. Unterstrasser, 2014 addressed this discrepancy and pinpointed the Naiman model to be an outlier model. Unterstrasser, 2016 performed a more thorough comparison of various LES models used for early contrail simulations. They found that the wake vortex descent and decay in the Naiman model is similar to that of the other models which implies that the

discrepancies are likely associated with microphysics. The present study tackles this problem and attempts to give an explanation of the observed behaviour. Generally I appreciate this effort, the format and the intention behind this study.

Often such tests are added as appendices in more comprehensive studies. I agree that often details matter which require a somewhat longer description than possible in such appendices. Given the rather short length of the manuscript, I recommend to extend the study by a few aspects to make it a more substantial scientific contribution that is suitable to be published as independent research.

The presented results leave many questions open and I get sometimes the impression that the presented results are not fully understood. Some results seem implausible. At best, they are not explained well enough. Moreover, I think that the evaluation of some relationships or quantities is not overly useful. In general, the text touches many aspects only superficially.

I recommend publication, only after including a more careful analysis, a more thoughtful presentation and a better explanation of your results.

**Major issues:**

A1. Motivating your work with the low level of scientific understanding by citing a reference to Penner et al, 1999 (collecting scientific results about 20 years old) is awkward. Science has progressed since. Sausen et al, 2005, Lee et al, 2010, Burkhardt and Kärcher, 2011, Bock and Burkhardt, 2016

A2. The results section looks more like a technical report where hard facts are listed. Implications and interpretation of the simulation results are only touched in a few cases. I could live with it, if I understood all your model results. However, this is not the case which is outlined in the following.

A3. Ice crystal formation in contrails is completed after one second or so. Hence, the fraction of surviving particles as you show in Fig. 2 should be a monotonically

decreasing function. It is not trustworthy when one simulation (red line) shows a late time increase after t=400s. So what's wrong, something in the model or in the post-processing tools?

A4. Several aspects of the evolution of the size distribution (SD) in Fig. 3 look peculiar.
1.First of all, your chosen colours for B0 and B2 are hard to distinguish. Please replace one.
2.Why do you change the y-range for $t \geq 60s$, when there is no need to?
3. The large droplet mode is controlled by $RH_i$. Hence for the right end of the SD, all solid lines are basically identical and similarly all dashed lines. For $t = 15, 30$ and $45$ s it looks like this may not be the case. Can you clarify this.
4. What really irritates me is the fact the left tail (the so called sublimation tail) develops faster for higher $RH_i$. How can this be? There's three times more water vapour available for deposition in the $RH_i = 130\%$-cases. So why should ice crystals be more susceptible to sublimation in this case? This result is counter-intuitive, implausible and different to all other models. Hence, it must be explained in detail, why your models behaves like this. As a consequence, the ice crystal number in Fig. 2 drops faster for higher $RH_i$. For the red case, even the final survival fraction is lower. This result is hard to believe. Did you carefully check all your model components?

A5. It is discouraging, if you mix up things and reviewers have to disentangle them: The terminology of your simulations is misleading. Runs B0, B1, L0 and L1 use a temperature-dependent $a_k$ which are compared to runs with constant $a_k = 1 \times 10^{-9}$ m. Your presentation implies that including the temperature dependence makes a large difference. However, the temperature dependence itself is not the reason for the observed differences (the temperature dependence is anyway weak, as you shows in Table 1 and Fig. 1). It is simply that the $a_k$-value is about 2.3 times higher if you use your temperature-dependent expression. So you basically compare cases with $a_k = 1 \times 10^{-9}$ m and $a_k = 2.3 \times 10^{-9}$ m.
By the way, why do list $a_k$-values for three different $\sigma$-values, if only one case is used

in the simulations?

p4. l.9-10: "results in substantially lower Kelvin correction for smaller particles". This is misleading as the correction factor is constant over the whole radius range, only the absolute change is larger for smaller particles.

**Major to Minor issues:**

B1. The paragraph (p.3 l.18- l.20) sounds like a perfect motivation to carry out sensitivity simulations varying the initial size distribution. If the true initial size distribution is not known, a model offers the unique opportunity to vary this parameters. This is particularly interesting in this study. The Kelvin effect has a prominent effect on the shape of the size distribution as you show in Fig. 3. So a variation of the initial size distribution is directly relevant to the main aspect of your paper. This may be also a reason for discrepancies between models.

B2. Similarly, I recommend to carry out the L4 and L5 simulations. You say, those simulations are not necessary, as Picot et al, 2015 showed that no crystal loss occurs. One main motivation of your work was the discrepancy between the various models. So in this sense, referring to another model is a bit contradictory. It would be interesting to know, if your model behaves similarly.

B3. To me it is unclear, what you want to demonstrate with the bottom row of Fig. 2. Mean particle size is mainly controlled by growth of detrained ice crystals being outside of the vortex system. The crystal loss, on the other hand, occurs inside the vortex system. For me it makes no sense to link those to quantities, as they are not really physically connected. I recommend to remove the paragraph from p.4 l.32 to p.5 l.5 and the sentence in the abstract/conclusions.

B4. Personally, I think that analyses of optical depth of young contrails are not overly useful, as this quantity is linked to radiation and climate aspects. LES of young contrails are not directly relevant to this. For this, contrails must be simulated over a much longer time (at least several hours). Optical depth decreases substantially over the first half an

hour, as the contrail gets usually tilted by vertical wind shear (a process absent in your simulations). So the given optical depth values are only a snapshot. The differences you find may not be long-lasting. Indeed, Unterstrasser et al, 2016 presents contrail-cirrus simulations over 6 hours and switching off the Kelvin effect had barely an effect on contrail properties (all simulations were initialised with the same 5 min old contrail, though)

B5. A point-to-point comparison between various models as done on p5 l21 is not leading anywhere. Contrail optical depth depends on a multitude of parameters. So you always find simulations with similar, yet not identical parameters which leave enough room for arguing that for this or that reason the optical depth is similar or smaller/larger in the one case. Unterstrasser, 2016 presents a more rigorous evaluation exercise that accounts for the multi-parameter nature of the problem and that is also able to disclose systematic model differences as mentioned in the introduction of this review.

B6. Naiman speculated that they might have used too few computational particles and that this could have led to the discrepancy with other LES models. How many particles did you use? Do your results depend on this numerical parameter? Did you check convergence of your results?

**Minor Issues:**

C1. I don't want to downplay the possible effect the early temperature difference by including/excluding the exhaust enthalpy has on contrail properties. Nevertheless, it is noteworthy that after 100s the excess plume temperature is not affected at all by this model aspect.

C2. I recommend to split Fig. 1 for clarity reasons. The left column shows LES results and the right column shows simple physical relations without a connection to your LES results.

C3. You cite several Inamdar papers from the recent past. I am not sure, if all those

are peer-reviewed contributions. If not, I recommend to reduce references to them and instead repeat the results in this study. For example, p.6, l.9/10 cites an important result of your recent work. Has it gone through peer review?

C4. Can you add the expression for $\sigma$? Do you vary it independently of T? The legend of Fig. 1 right alludes to this.

C5. p.2 l.27: What do you want to say here? Can you make a clearer connection between the availability of measurement data and what you say in the rest of the sentence.

C6. p.3 l.33: The plume temperature is constant!? I do not understand this statement. The plume temperature increases due to adiabatic heating. It may help if you describe in more detail how you compute the excess temperature. How is your reference temperature determined?

C7. p.6 l.4: Be more specific about how measurements can help. Otherwise this statement is pointless.

**Technical points:**

T1. Many author names are mis-spelled (often missing german umlauts or french accents): Sölch, Görsch, Kühnlein, Kärcher, Helie, Nybelen, Schäuble, peter J. Newton to name only a few!

T2. For units the regular font is usually used.

T3. # is no SI unit. I guess you can just remove it.

T4. The numerical treatment of the Kelvin effect in the Sölch Kärcher model is described in more detail in Unterstrasser et al, 2016 and can be cited for reference.

T5. The axis annotations and legends are too small in many plots. In Fig. 1, the legend misses the unit m for $a_k$. In Fig. 3 it suffices to show the y-axis on the left column. Inserting the time label in each plot would save a lot space.

T6. p.3 l.3: better write $\Delta x_{jet}$ and $\Delta x_{vort}$ separately.

T7. p.2 l.3 RHi is clearly not ice supersaturation.

**References:**

L. Bock und U. Burkhardt. Reassessing properties and radiative forcing of contrail cirrus using a climate model. J. Geophys. Res., 121(16):9717-9736, 2016. doi: 10.1002/2016JD025112

U. Burkhardt und B. Kärcher. Global radiative forcing from contrail cirrus. Nature Clim. Ch., 1(1):54-58, 2011

D. Lee, G. Pitari, V. Grewe, K. Gierens, J. Penner, A. Petzold, M. Prather, U. Schumann, A. Bais, T. Berntsen, D. Iachetti, L. Lim, und R. Sausen. Transport impacts on atmosphere and climate: Aviation. Atmos. Environ., 44(37):4678 - 4734, 2010.

R. Sausen, I. Isaksen, V. Grewe, D. Hauglustaine, D. Lee, G. Myhre, M. Köhler, G. Pitari, U. Schumann, F. Stordal, et al. Aviation radiative forcing in 2000: An update on IPCC (1999). Meteorol. Z., 14(4):555-561, 2005

S. Unterstrasser. Properties of young contrails - a parametrisation based on large-eddy simulations. Atmos. Chem. Phys., 16(4):2059-2082, 2016. doi: 10.5194/acp-16-2059-2016

S. Unterstrasser, K. Gierens, I. Sölch, und M. Lainer. Numerical simulations of homogeneously nucleated natural cirrus and contrail-cirrus. Part 1: How different are they? Meteorol. Z., 2016a. doi: 10.1127/metz/2016/0777
* * *

---

## Author Comment (AC2) · 23 Dec 2016

**Interactive comment* on "Sensitivity of particle loss to the Kelvin effect in LES of young contrails" by Aniket R. Inamdar et al.**

Aniket R. Inamdar[1], Alexander D. Naiman[2], Sanjiva K. Lele[1,2], and Mark Z. Jacobson[3]

[1]Dept. of Mech. Engg.
[2]Dept. of Aero. & Astro.
[3]Dept. of Civil & Env. Engg.
[1,2,3]Stanford University, California, USA

*Correspondence to:* Aniket R. Inamdar (`ainamdar@stanford.edu`)

The authors would like to thank Anonymous Referee #2 for his/her comments. The following are the authors' responses to the comments made by the referee:

*Comment*:

5     **The study presents LES of young contrails and focuses mainly on one aspect, i.e. the treatment of the Kelvin effect in the ice microphysical model (which is also reflected in the title). The numerical model has been used before in the study of Naiman et al, 2011. Compared to other LES results, the latter study showed a rather weak sensitivity to ambient relative humidity which appears counter-intuitive. Unterstrasser, 2014 addressed this discrepancy and pinpointed the Naiman model to be an outlier model. Unterstrasser, 2016 performed a more thorough comparison of various LES models used for early contrail simulations. They found that the wake vortex descent and decay in the Naiman model**
10     **is similar to that of the other models which implies that the discrepancies are likely associated with microphysics. The present study tackles this problem and attempts to give an explanation of the observed behaviour. Generally I appreciate this effort, the format and the intention behind this study.**

*Response*:

    We thank the referee for this comment. It is indeed important to find the source of discrepancy between various models.
15  Unterstrasser (2014) pointed out the fact that results from CDP_IF2 and EULAG-LCM do not agree and did not point out the cause of the discrepancy. There the author references observed RHi-dependent onset of crystal loss in his own simulations and cites a possible explanation for ice mass loss (not crystal loss) from earlier Bulk-Microphysics simulations (Unterstrasser et al., 2008) (Sec. 5.1 therein) to claim "implausibility" of the results in Naiman et al. (2011). It is true that no two of the four models {Naiman et al. (2011); Unterstrasser (2014); Picot et al. (2015); Huebsch and Lewellen (June 5 to 8, 2006, San
20  Francisco, CA, USA) } can reproduce each other's sensitivities exactly. Hence, classifying Naiman et al. (2011) as a "outlier" may be unwarranted. A closer look at the analysis of crystal loss considered in Unterstrasser (2016) reveals that the choice of the functional form of $\hat{a}(x)$ in Eqn. (9) therein seems to be an assumption made based on the data generated by EULAG-LCM. In making this choice, the implicit assumption is that $EI_{soot}$ is less significant to crystal loss than RHi ($EI_{soot}$-dependence is introduced adhoc in coefficients to $z_{atm}$ and $z_{emit}$ that are not related to $EI_{soot}$) . Then parameter values are inferred from
25  data generated using EULAG-LCM (not in-situ or experimental data) and a claimed non-conformity to this curve-fit model is

given as proof of Naiman et al. (2011)'s "outlier" behavior. This approach does not seem to provide sufficient information to determine whether an issue exists with the model. For example, what are the uncertainties in the parameter estimates in their Eqn. 10(a-h)? What are the resulting $z_\Delta$ point-wise (or bin-wise) 95% confidence intervals to the curve fit (similar to the black dotted lines in our Fig. 2(Left) in the revised manuscript, a global L-2 error is not sufficient to judge efficiency of the curve-fit over the entire range of $z_\Delta$)? How do the models fare compared to the (presumably large) confidence intervals (77 data points to find minima of error and bias in a 6-D space)?

We have shown in our manuscript the major cause of discrepancy between models employing Lagrangian Particle Tracking(LPT) (Paoli and Shariff, 2016) without making claims of model errors or biases.

*Comment*:

**Often such tests are added as appendices in more comprehensive studies. I agree that often details matter which require a somewhat longer description than possible in such appendices. Given the rather short length of the manuscript, I recommend to extend the study by a few aspects to make it a more substantial scientific contribution that is suitable to be published as independent research.**

**The presented results leave many questions open and I get sometimes the impression that the presented results are not fully understood. Some results seem implausible. At best, they are not explained well enough. Moreover, I think that the evaluation of some relationships or quantities is not overly useful. In general, the text touches many aspects only superficially.**

**I recommend publication, only after including a more careful analysis, a more thoughtful presentation and a better explanation of your results.**

*Response*:

We intend for this manuscript to be concise and convey one major scientific finding - evaporative loss of particles in early persistent contrails is strongly sensitive to modeling of the Kelvin effect and this is the main source of discrepancy between some existing LES models. Many parametric models for contrails designed to be employed in larger GCMs use LES data from these discrepant LES models to infer parameter values (*e.g.* Unterstrasser (2016)). Some studies use LES data for young contrails to initialize simulations over longer time horizons (Unterstrasser et al., 2016; Jacobson et al., 2011). Such studies stand to directly benefit from our analysis. This is mentioned in the revised manuscript at p2.l16. We have included in the manuscript some of the most relevant explanations from the responses to the referee's comments given below so as to make the manuscript thorough in its analysis of the impact of the Kelvin effect.

**Major Comments:**

*Comment*:

**A1: Motivating your work with the low level of scientific understanding by citing a reference to Penner et al, 1999 (collecting scientific results about 20 years old) is awkward. Science has progressed since. Sausen et al, 2005, Lee et al, 2010, Burkhardt and Karcher, 2011, Bock and Burkhardt, 2016**

*Response*:

Newer studies have now been appropriately cited in the Introduction of the revised manuscript (at p1.l11-16).

*Comment*:

**A2: The results section looks more like a technical report where hard facts are listed. Implications and interpretation of the simulation results are only touched in a few cases. I could live with it, if I understood all your model results. However, this is not the case which is outlined in the following.**

*Response*:

In the following responses, we have clarified the doubts raised by the referee. The implications of our results are concisely stated in the Conclusions section of the manuscript. Admittedly, we may have cut short on the discussion of our results in our attempt to keep the manuscript concise. Important clarifications sought by the referee are now included in the Results section of the revised manuscript.

**A3: Ice crystal formation in contrails is completed after one second or so. Hence, the fraction of surviving particles as you show in Fig. 2 should be a monotonically decreasing function. It is not trustworthy when one simulation (red line) shows a late time increase after t=400s. So what's wrong, something in the model or in the post- processing tools?**

*Response*:

Both the model and the post-processing tools are thoroughly vetted and we were unable to find any errors. Indeed, we observe an increase in the survival rate after $t \approx 400$s for high $EI_{soot}$ and no Kelvin correction. Monotonicity is not an intrinsic property of the survival rate but a mere consequence of modeling assumptions. All particles reduced to the bare soot core ($r = 15$nm) are considered lost, but not deactivated for vapor deposition. In the absence of the Kelvin effect, crystal loss during the vortex phase is primarily due to plume being warmer than the ambient with additional aid from turbulent fluctuations (Kärcher et al., 2014). After the vortex phase, as the plume sloshes due to buoyancy and mixes with the ambient, its temperature relaxes to that of the local ambient (see Fig. 1(Left) in the revised manuscript) and RHi begins to rise. Now, these still activated soot cores can resume uptake of the available excess vapor. This is what is observed for the red curve in Fig. 3 (a, Top) of the revised manuscript. Cases with Kelvin effect are not seen to have this behavior as the lost particles never overcome the Kelvin barrier. Lewellen (2012) theoretically shows monotonicity of the particle survival rate only for cases with Kelvin effect.

Schumann (1996) notes that for high temperatures as seen in the jet phase, most soot activation occurs through "condensation freezing mode" (Pruppacher and Klett, 2010). However, for low temperatures (as seen during the buoyant sloshing and dispersion phases), ice nuclei can be activated directly for vapor deposition (Pruppacher and Klett, 2010). Bailey and Hallett (2004, 2002) observe that $RHi < 125\%$ is sufficient to activate particles for temperatures less than $-42^{o}C$. Even at the high temperatures observed during the jet phase, Kärcher et al. (2015) notes that in-situ measurements do not rule out direct vapor activation. In our model we assume that for low temperatures, soot is activated directly for vapor deposition (Paoli et al., 2004) while Picot et al. (2015) sets the higher bar of condensation freezing even during the dispersion phase. Noting that $p_{sat}^{water}(T = 219K) \approx 3.5Pa$ and $p_{sat}^{ice}(T = 219K) \approx 2.1Pa$, for the RHi values considered in Picot et al. (2015), this means that a particle that is once reduced to its soot core is never reactivated - even in their no Kelvin cases.

It is pertinent to note that this difference in the model assumptions causes a mere ∼4% increase in the survival rate of our red curve in Fig. 3(a, Top) of the revised manuscript - the no Kelvin case that is anyway unphysical. We have noted this in the revised manuscript at p3.l21-24, p5.l20-26

*Comment*:

**A4: Several aspects of the evolution of the size distribution (SD) in Fig. 3 look peculiar.**

**1.First of all, your chosen colours for B0 and B2 are hard to distinguish. Please replace one.**

*Response*:

B2, B3 curves are now green while L0, L1 curves are now blue. The contrasting colors in each panel are now clear and easily distinguishable.

**2.Why do you change the y-range for t ? 60s, when there is no need to?**

*Response*:

The purpose of changing the y-range is simply to make sure that the figure shows only the relevant range and reduce unnecessary white space.

**3. The large droplet mode is controlled by RHi. Hence for the right end of the SD, all solid lines are basically identical and similarly all dashed lines. For t = 15, 30 and 45 s it looks like this may not be the case. Can you clarify this.**

*Response*:

The plume RHi remains the same irrespective of the Kelvin effect modeling. So, the total uptake of available vapor for ice growth is the same, however its distribution between small and large particles is different based on how strong the Kelvin effect is. The near 100% plume RHi (Naiman et al., 2011; Paoli and Shariff, 2016) suggests that almost all available excess vapor in the plume volume is used up by the ice particles. We see the ice mass in the plume is indifferent to the Kelvin effect in Fig. 1(Right) of the revised manuscript. So, if $a_k = 0$, some of the total excess vapor budget is accounted for by the abundant small particles leaving less for the large particles and hence reducing their prevalence, *i.e.* when Kelvin effect is lowered or suppressed, the vapor uptake by small particles results in the reduction in availability of vapor for, and thus, the prevalence of large particles, albeit marginally. This is what is seen in the SDs at 90s,120s and 300s (we presume these are the times the referee meant as the SD is almost identical for large sizes at 15s, 30s, and 45s). This has been clarified in the revised manuscript at p6.l16-21.

**4. What really irritates me is the fact the left tail (the so called sublimation tail) develops faster for higher RHi. How can this be? There's three times more water vapour available for deposition in the RHi = 130%-cases. So why should ice crystals be more susceptible to sublimation in this case? This result is counter-intuitive, implausible and different to all other models. Hence, it must be explained in detail, why your models behaves like this. As a consequence, the ice crystal number in Fig. 2 drops faster for higher RHi. For the red case, even the final survival fraction is lower. This result is hard to believe. Did you carefully check all your model components?**

*Response*:

We believe all model components have been implemented accurately. The slower "spectral ripening" of the size PDF for low RHi is explained below.

Let $N_{tot}(t)$ be the total number of ice crystals at time $t$. Until the onset of particle loss, the survival rate is near 100%. So, $N_{tot}(t) = N_{tot} = const.$. Let $N(r,t)$ represent the number of ice particles of size $\leq r$ at time $t$. Thus, the size PDF is defined as $n(r,t) = \frac{1}{N_{tot}} \frac{\partial N(r,t)}{\partial r}$. It follows that,

$$\bar{r}(t) = \int_{r_{low}}^{r_{hi}} r n(r,t) dr \qquad AND \qquad \overline{r^2}(t) = \int_{r_{low}}^{r_{hi}} r^2 n(r,t) dr$$

5  $\quad \sigma^2(t) = \int_{r_{low}}^{r_{hi}} (r - \bar{r})^2 n(r,t) dr = \overline{r^2} - \bar{r}^2$ \hfill (AR 1)

where $\sigma^2(t)$ is the variance of the size PDF and a measure of its width. Until onset of crystal loss, the mean particle size grows very slowly (see Fig. 3(a, Center) of revised manuscript) so we have $\frac{d\bar{r}}{dt} \approx 0$. Hence, we expect $\frac{d\sigma^2}{dt} \approx \frac{d\overline{r^2}}{dt}$. Using Naiman et al. (2011) (their Eqn. 9), we thus have,

$$\frac{d\sigma^2}{dt} \approx \frac{d\overline{r^2}}{dt} = \frac{d}{dt} \frac{\sum_{i=1}^{N_{tot}} r^2(t)}{N_{tot}} \quad | \quad \text{Eulerian to Lagrangian frame as Lagrangian resolution is}$$

10 \hfill converged (Naiman, 2011)(Appendix A therein)

$$= \frac{\sum_{i=1}^{N_{tot}} \frac{dr^2(t)}{dt}}{N_{tot}} = 2 \frac{D_v}{\rho_{ice}} \frac{\sum_{i=1}^{N_{tot}} \Delta\rho_{excess}^{vapor}(t)}{N_{tot}} \quad | \quad r_{low} \geq 2 \times 10^{-7} m \therefore \Phi(\frac{a_k}{r(t)}) \approx 1 \text{ and } \Delta\rho_{excess}^{vapor} \neq f(r)$$

$$\therefore \frac{d\sigma^2}{dt} \propto \overline{\Delta\rho_{vapor}^{excess}}$$ \hfill (AR 2)

The smallest particles are large enough to be indifferent to the Kelvin effect (99.98% of the particles are larger than $2 \times 10^{-7}$m up to $t \leq 20$s for $RHi = 130\%$ and $t \leq 35$s for $RHi = 110\%$). So, $\Delta\rho_{excess}^{vapor}$ is only a weak function of $r$ and $\overline{\Delta\rho_{excess}^{vapor}}$ is

15 driven by the temperature experienced by the particles (up to the aforementioned times). This is sufficient to reason the faster spreading of the size PDF in high RHi cases - higher RHi $\Rightarrow$ higher $\overline{\Delta\rho_{vapor}^{excess}}$ $\Rightarrow$ faster growth of $\sigma^2(t)$. Thus, the size PDF of cases with higher RHi starts having small particles (susceptible to Kelvin loss) earlier ($\sim$ 20s for $RHi = 130\%$ and $\sim$ 35s for $RHi = 110\%$).

In cases with Kelvin effect, once particles appear in size bins affected by the Kelvin effect (*i.e.* $r_{bin} \leq 2 \times 10^{-7}$), the assump-

20 tions of the above analysis are no longer valid. Now, cases with higher $a_k$ loose particles faster due to apparent subsaturation experienced by them and consequently, the sublimation tail spreads faster for higher $a_k$ values. Simultaneously, we see a rise in mean particle size $\bar{r} = \frac{\sum_{i=1}^{N_{tot}(t)} r_i(t)}{N_{tot}(t)}$ as $N_{tot}$ begins to fall while the total mass uptake is the same (given an RHi) for all $a_k$ values. Eventually the size PDF evolves in a self-similar manner (Lewellen, 2012) (their "slow growth regime"). However, for $a_k = 0$, the above analysis continues to hold until the smallest size bin (or the bin with lost particles, $r_{bin} = 15$nm) gets

25 populated (this happens earlier for $RHi = 130\%$). Thus, case B4 sees early onset of particle loss than B5. As explained earlier, this "loss" in the case of no Kelvin correction is only temporary as after buoyant sloshing of the warm plume and mixing with the cool ambient, these activated soot cores regrow.

In Fig. AR 1 we have shown the evolution of $\sigma^2$ for high $EI_{soot}$ cases. We see that for a given RHi, the evolution of $\sigma^2$ is identical up to $t = 20$s for $RHi = 130\%$ and $t = 35$s for $RHi = 110\%$ and that $\sigma^2$ grows faster for higher RHi. Beyond these times, the cases with higher values of $a_k$ see faster growth of $\sigma^2$.

[Figure]

**Figure AR 1.** Evolution of width of size PDF for RHi 130%(solid lines) and 110%(dashed lines) and $a_k = 0$m(red), $a_k = 10^{-9}$m(green) and $a_k = 2.3 \times 10^{-9}$m(black)

This discussion is now included in the revised manuscript in Appendix B. As to why other models do not observe this behavior, we will refrain from speculating as we are not privy to the specifics of their implementation.

*Comment*:

**A5 - 1: It is discouraging, if you mix up things and reviewers have to disentangle them: The terminology of your simulations is misleading. Runs B0, B1, L0 and L1 use a temperature-dependent $a_k$ which are compared to runs with constant $a_k = 1 \times 10^{-9}$ m. Your presentation implies that including the temperature dependence makes a large difference. However, the temperature dependence itself is not the reason for the observed differences (the temperature dependence is anyway weak, as you shows in Table 1 and Fig. 1). It is simply that the $a_k$-value is about 2.3 times higher if you use your temperature-dependent expression. So you basically compare cases with $a_k = 1 \times 10^{-9}$ m and $a_k = 2.3 \times 10^{-9}$ m.**

*Response*:

Using $a_k = 10^{-9}$ m is an order-of-magnitude accounting of the Kelvin effect . For $a_k$ to be $10^{-9}$m as used in Picot et al. (2015) , the $\sigma$ would have to be $\sim 0.047$Jm$^{-2}$ which is unphysically low for the temperatures relevant to development of persistent contrails ($T = 218$K, say) (Pruppacher and Klett, 2010). Using a physically plausible, temperature-dependent value

of $\sigma \approx 0.107 \text{Jm}^{-2}$ the $a_k$ value is obtained is $\approx 2.3 \times 10^{-9}$m. Indeed, this means that we fix for B0, B1, L0, L1, U0 and U1, $a_k = 2.3 \times 10^{-9}$m. The usage of the term "temperature-dependent" may be misleading and this has been removed in the revised manuscript.

*Comment*:

5     **A5 - 2: By the way, why do list $a_k$-values for three different $\sigma$-values, if only one case is used**

*Response*:

The purpose of listing $a_k$ values for 9 different $(\sigma, T)$ combinations, is to show that for a realistic range of $\sigma$ and $T$, the value of $a_k$ is still $\sim 2 \times 10^{-9}$. The dotted black lines in Fig. 2(Left) of the revised manuscript (that use extreme values of $a_k$ from our Table 1) then stress that a realistic variation in $a_k$ due to changes in $(\sigma, T)$ does not affect the Kelvin correction

10 dramatically. However, an order-of-magnitude value of $a_k$ is seen to significantly change the correction factor.

*Comment*:

**A5 - 3: p4. l.9-10: "results in substantially lower Kelvin correction for smaller particles". This is misleading as the correction factor is constant over the whole radius range, only the absolute change is larger for smaller particles.**

*Response*:

15     The Kelvin correction factor $\Phi = \exp\left(\frac{a_k}{r}\right)$ is indeed a function of radius! So, the value of the parameter $a_k$ may be constant, but the correction factor experienced by different radii is certainly different - which is shown in Fig. 2(Left) of the revised manuscript.

**Major to Minor Comment:**

*Comment*:

20     **B1: The paragraph (p.3 l.18- l.20) sounds like a perfect motivation to carry out sensitivity simulations varying the initial size distribution. If the true initial size distribution is not known, a model offers the unique opportunity to vary this parameters. This is particularly interesting in this study. The Kelvin effect has a prominent effect on the shape of the size distribution as you show in Fig. 3. So a variation of the initial size distribution is directly relevant to the main aspect of your paper. This may be also a reason for discrepancies between models.**

25     *Response*:

Thank you for this comment. The major discrepancies - low RHi sensitivity and correspondingly heightened $EI_{soot}$ sensitivity - are already adequately accounted for by the different Kelvin effect treatment (Please see p5.l9-12, p6.l28-34 in the revised manuscript). Besides marginally reducing the time to onset of particle loss, changing the initial size PDF is expected to not have any significant impact. In our above analysis, $\sigma^2(0) = 0$, so evolution of a thicker initial size PDF (*i.e.* $\sigma^2(0) = \sigma_0^2 > 0$)

30 will put particles in $r \leq 2 \times 10^{-7}$m range earlier to start particle loss earlier (also see Fig. 6 of Unterstrasser and Sölch (2010)). This has been noted in Appendix B of the revised manuscript.

*Comment*:

**B2: Similarly, I recommend to carry out the L4 and L5 simulations. You say, those simulations are not necessary, as Picot et al, 2015 showed that no crystal loss occurs. One main motivation of your work was the discrepancy between the various models. So in this sense, referring to another model is a bit contradictory. It would be interesting to know, if your model behaves similarly.**

*Response*:

There is negligible discrepancy between our low Kelvin effect case for low $EI_{soot}$ and Picot et al. (2015). So, at least ex-post, Fig. 3(b) in the revised manuscript vindicate our choice of not choosing to run cases L4, L5.

*Comment*:

**B3: To me it is unclear, what you want to demonstrate with the bottom row of Fig. 2. Mean particle size is mainly controlled by growth of detrained ice crystals being outside of the vortex system. The crystal loss, on the other hand, occurs inside the vortex system. For me it makes no sense to link those to quantities, as they are not really physically connected. I recommend to remove the paragraph from p.4 l.32 to p.5 l.5 and the sentence in the abstract/conclusions.**

*Response*:

The argument about exponential relationship between size and survival rate is being made only for the vortex phase(our p5.l33 in the revised manuscript *...we observe an exponential relation between survival rate and size during the vortex phase*). As seen in Inamdar et al. (2016) (their Fig. 7) and also Unterstrasser (2014)(first panel of their Fig. 1), there is hardly any secondary curtain when the coherent vortical structures are descending through stratified ambient - thus size and survival rate are certainly physically linked up to the end of the vortex phase and it is logical to seek a relation between them. This observation will help in modeling the size and survival rate in any model for contrail evolution up to the vortex phase. As we argue in our manuscript (p5.l35 of revised manuscript *But, as the plume temperature...*), this becomes progressively invalid after the coherent vortical structures are destroyed and the plume sloshes and mixes with the ambient while its temperature falls. In Fig. 3(a, Bottom) of the revised manuscript, we see that for a given $a_k$, the plot for both high and low RHi is linear and has similar slopes up to the destruction of the coherent vortex system. After that, the curves for different RHi diverge and we may no longer seek an exponential relation beyond this point. In the revised manuscript we stress that this exponential relationship is valid only during the vortex phase at p6.l1.

*Comment*:

**B4: Personally, I think that analyses of optical depth of young contrails are not overly useful, as this quantity is linked to radiation and climate aspects. LES of young contrails are not directly relevant to this. For this, contrails must be simulated over a much longer time (at least several hours). Optical depth decreases substantially over the first half an hour, as the contrail gets usually tilted by vertical wind shear (a process absent in your simulations). So the given optical depth values are only a snapshot. The differences you find may not be long-lasting. Indeed, Unterstrasser et al, 2016 presents contrail- cirrus simulations over 6 hours and switching off the Kelvin effect had barely an effect on contrail properties (all simulations were initialised with the same 5 min old contrail, though)**

*Response*:

The referee points out that Unterstrasser et al. (2016) observe little sensitivity of contrail optical properties to the Kelvin effect past 5 mins., given a particular initialization (that was inferred from early contrail simulation which assumed one value of $a_k$). However, it is pertinent to ask - "How sensitive are these long time horizon simulations to different initializations?". So, if the same simulations were initialized using our B2 case vs. our B0 case, how different would the long-time predictions be? Extinction is proportional to number of ice particles and we see by 5min. our B2 case to have $\sim 200\%$ higher number of ice particles as compared to our B0 case. This has been noted at p5.l13-19 in the revised manuscript.

Indeed, the OD values for young contrails are not directly useful for studying radiation and climate aspects. However the objective here is to see if we produce reasonable values of OD and extinction compared to other models employed in similar simulations for similar time horizons (Picot et al., 2015).

*Comment*:

**B5: A point-to-point comparison between various models as done on p5 l21 is not leading anywhere. Contrail optical depth depends on a multitude of parameters. So you always find simulations with similar, yet not identical parameters which leave enough room for arguing that for this or that reason the optical depth is similar or smaller/larger in the one case. Unterstrasser, 2016 presents a more rigorous evaluation exercise that accounts for the multi-parameter nature of the problem and that is also able to disclose systematic model differences as mentioned in the introduction of this review.**

*Response*:

As stated before, the objective here is merely to show that the model produces OD values that are "reasonable" and not "outliers". This is not uncommon in literature (Picot et al., 2015; Naiman et al., 2011; Paugam et al., 2010). We address the comment about Unterstrasser (2016) in our response to the referee's first comment.

*Comment*:

**B6: Naiman speculated that they might have used too few computational particles and that this could have led to the discrepancy with other LES models. How many particles did you use? Do your results depend on this numerical parameter? Did you check convergence of your results?**

*Response*:

We use a total of $8 \times 10^6$ simulation particles(SIPs)/numerical particles(NPs). Please see Appendix A of Naiman (2011) for a detailed analysis of sensitivity to this parameter. We see that the results are converged for this SIP resolution.

**Minor Comment:**

*Comment*:

**C1: I don't want to downplay the possible effect the early temperature difference by including/excluding the exhaust enthalpy has on contrail properties. Nevertheless, it is noteworthy that after 100s the excess plume temperature is not affected at all by this model aspect.**

*Response*:

This is exactly our finding in Conclusion # 4.

*Comment*:

**C2: I recommend to split Fig. 1 for clarity reasons. The left column shows LES results and the right column shows simple physical relations without a connection to your LES results.**

*Response*

This has been done in the revised manuscript.

*Comment*:

**C3: You cite several Inamdar papers from the recent past. I am not sure, if all those are peer-reviewed contributions. If not, I recommend to reduce references to them and instead repeat the results in this study. For example, p.6, l.9/10 cites an important result of your recent work. Has it gone through peer review?**

*Response*:

These are conference papers and are publicly available. Results from these papers are being compiled into Inamdar et al. (in progress, 2016b, i). Citing conference papers is not uncommon in the field, *e.g* Huebsch and Lewellen (June 5 to 8, 2006, San Francisco, CA, USA) is referenced widely in contrail LES literature.

*Comment*:

**C4: Can you add the expression for $\sigma$? Do you vary it independently of T? The legend of Fig. 1 right alludes to this.**

*Response*:

As stated in the manuscript, the value of $\sigma$ is held constant for "temperature dependent" cases at $0.107 \mathrm{Jm}^{-2}$.

*Comment*:

**C5: p.2 l.27: What do you want to say here? Can you make a clearer connection between the availability of measurement data and what you say in the rest of the sentence.**

*Response*:

Febvre et al. (2009) reports measurements for $\sim 2.5$min. old contrails, Gayet et al. (2012) reports measurements for 70s, 105s, 205s old contrails. We wished to see if the impact of exhaust enthalpy subsides by $\sim 100$s or else the exclusion of it may have affected our ability to compare simulation data with observations made at those time horizons. As it turns out, exhaust enthalpy will not affect this comparison. This has been elaborated in the revised manuscript at p3.l5-9.

*Comment*:

**C6: p.3 l.33: The plume temperature is constant!? I do not understand this statement. The plume temperature increases due to adiabatic heating. It may help if you describe in more detail how you compute the excess temperature. How is your reference temperature determined?**

*Response*:

The plume "excess" temperature is constant. Thank you for pointing out this typo! The definition given below is also included in the revised manuscript:

$$\Delta T_{plume} = \frac{\int_{\Omega} \left( T - T_0 - \left( \frac{dT}{dy} \right)_{N_{bv}=0.01s^{-1}} (y - y_0) \right) dV}{\int_{\Omega} dV} \tag{AR 3}$$

where $\Omega$ is the plume volume defined in the manuscript.

Comment:

**C7. p.6 l.4: Be more specific about how measurements can help. Otherwise this statement is pointless.**

*Response*:

Please see revised manuscript at p7.l9-13.

**Technical Comments:**

Technical corrections T1-T3 and T5-T7 have been made in the revised manuscript.

*Comment*:

**T4: The numerical treatment of the Kelvin effect in the Sölch Kärcher model is described in more detail in Unterstrasser et al, 2016 and can be cited for reference.**

*Response*:

The referee cites Unterstrasser et al. (2016) to see details of the Kelvin effect implementation in Sölch and Kärcher (2010). However, it does not state what value of $a_k$ has been used in EULAG-LCM simulations to date - only that in this particular paper $a_k = 2.3e - 9$m and has been varied later from 0 to $4.6e - 9$m. Our results for high $EI_{soot}$ and $a_k = 1e - 9$m match very well with their previous simulations (Unterstrasser, 2014) (their "Gaussian" cases) and cases with low $EI_{soot}$ and $a_k = 1e-9$m compare well with Picot et al. (2015). A comparison of EULAG-LCM and NTMIX is done in Unterstrasser et al. (2014). So, we suspect $a_k = 1e - 9$m is the value used in earlier EULAG-LCM simulations.

**References**

Bailey, M. and Hallett, J.: Nucleation Effects on the habit of vapor grown ice crystals from -18 to -42C, Quaterly Journal of the Royal Meteorological Society, 128, 1461–1483, 2002.

Bailey, M. and Hallett, J.: Growth Rates and Habits of Ice Crystals between -20C and -70C, Journal of the Atmospheric Sciences, 61, 514–544, 2004.

Febvre, G., Gayet, J.-F., Minikin, A., Schlager, H., Shcherbakov, V., Jourdan, O., Busen, R., Fiebig, M., Kärcher, B., and Schumann, U.: On Optical and Microphysical Characteristics of Contrails and Cirrus, Journal of Geophysical Research, 114, D02204, 2009.

Gayet, J.-F., Shcherbakov, V., Voigt, C., Schumann, U., Schäuble, D., Jeßberger, P., Petzold, A., Minikin, A., Schlager, H., Dubovik, O., and Lapyonok, T.: The evolution of microphysical and optical properties of an A380 contrail in the vortex phase, Atmospheric Chemistry and Physics, 12, 6629–6643, 2012.

Huebsch, W. W. and Lewellen, D. C.: Sensitivity Study on Contrail Evolution, in: 36th AIAA Fluid Dynamics Conference and Exhibit, AIAA 2006-3749, June 5 to 8, 2006, San Francisco, CA, USA.

Inamdar, A. R., Lele, S. K., and Jacobson, M. Z.: Reduced Order Modeling of Contrails: Jet Induction and Vortex Phases, in: 8th AIAA Atmospheric and Space Environments Conference, AIAA Aviation, AIAA 2016-3136, 2016.

Inamdar, A. R., Lele, S. K., and Jacobson, M. Z.: Sensitivity of Early Persistent Contrails to Aircraft Type, Ambient Conditions and Micro-physical Modeling, Part 2: ODE Model, Atmospheric Chemistry and Physics, in progress, 2016a.

Inamdar, A. R., Naiman, A. D., Lele, S. K., and Jacobson, M. Z.: Sensitivity of Early Persistent Contrails to Aircraft Type, Ambient Conditions and Microphysical Modeling, Part 1: LES Results, Atmospheric Chemistry and Physics, in progress, 2016b.

Jacobson, M. Z., Wilkerson, J., Naiman, A., and Lele, S.: The effects of aircraft on climate and pollution. Part I: Numerical methods for treating the subgrid evolution of discrete size- and composition-resolved contrails from all commercial flights worldwide, Journal of Computational Physics, 2011.

Kärcher, B., Dörnbrack, A., and Sölch, I.: Supersaturation Variability and Cirrus Ice Crystal Size Distributions, Journal of the Atmospheric Sciences, 71, 2905–2926, 2014.

Kärcher, B., Burkhardt, U., Bier, A., Bock, L., and Ford, I.: The microphysical pathway to contrail formation, Journal of Geophysical Research, 120, 7893–7927, 2015.

Lewellen, D.: Analytic Solutions for Evolving Size Distributions of Spherical Crystals or Droplets Undergoing Diffusional Growth in Dif-ferent Regimes, Journal of the Atmospheric Sciences, 69, 417–434, 2012.

Naiman, A., Lele, S., and Jacobson, M. Z.: Large eddy simulations of contrail development: Sensitivity to initial and ambient conditions over first twenty minutes, Journal of Geophysical Research, 116, 2011.

Naiman, A. D.: Modeling Aircraft Contrails And Emission Plumes For Climate Impacts, Ph.D. thesis, Stanford University, 2011.

Paoli, R. and Shariff, K.: Contrail Modeling and Simulation, The Annual Reviews of Fluid Mechanics, 48, 393–427, 2016.

Paoli, R., Hélie, J., and Poinsot, T.: Contrail formation in aircraft wakes, Journal of Fluid Mechanics, 502, 361–373, 2004.

Paugam, R., Paoli, R., and Cariolle, D.: Influence of vortex dynamics and atmospheric turbulence on the early evolution of a contrail, Atmospheric Chemistry and Physics, 10, 3933–3952, 2010.

Picot, J., Paoli, R., Thouron, O., and Cariolle, D.: Large-eddy simulation of contrail evolution in the vortex phase and its interaction with atmospheric turbulence, Atmospheric Chemistry and Physics, 15, 7369–7389, 2015.

Pruppacher, H. R. and Klett, J. D.: Microphysics of Clouds and Precipitation, Springer, second edn., 2010.

Schumann, U.: On conditions for contrail formation from aircraft exhausts, Meteorologische Zeitschrift, 5, 4–23, 1996.

Sölch, I. and Kärcher, B.: A large-eddy model for cirrus clouds with explicit aerosol and ice microphysics and Lagrangian ice particle tracking, Quaterly Journal of the Royal Meteorological Society, 136, 2074–2093, 2010.

Unterstrasser, S.: Large-eddy simulation study of contrail microphysics and geometry during the vortex phase and consequences on contrail-to-cirrus transition, Journal of Geophysical Research: Atmospheres, 119, 7537–7555, 2014.

Unterstrasser, S.: Properties of young contrails – a parametrisation based on large eddy simulations, Atmospheric Chemistry and Physics, 16, 2059–2082, 2016.

Unterstrasser, S. and Sölch, I.: Study of contrail microphysics in the vortex phase with a Lagrangian particle tracking model, Atmospheric Chemistry and Physics, 10, 10 003–10 015, 2010.

Unterstrasser, S., Gierens, K., and Spichtinger, P.: The evolution of contrail microphysics in the vortex phase, Meteorologische Zeitschrift, 17, 145–156, 2008.

Unterstrasser, S., Paoli, R., Sölch, I., Kühnlein, C., and Gerz, T.: Dimension of aircraft exhaust plumes at cruise conditions: effect of wake vortices, Atmospheric Chemistry and Physics, 14, 2713–2733, 2014.

Unterstrasser, S., Gierens, K., Sölch, I., and Lainer, M.: Numerical simulations of homogeneously nucleated natural cirrus and contrail-cirrus. Part 1: How different are they?, Meteorologische Zeitschrift, Pre-Publication Article, 2016.